# Development of Livestock-Associated Methicillin-Resistant *Staphylococcus aureus* (LA-MRSA) Loads in Pigs and Pig Stables During the Fattening Period

**DOI:** 10.3390/vetsci11110558

**Published:** 2024-11-11

**Authors:** Karl Pedersen, Martin Weiss Nielsen, Mette Ely Fertner, Carmen Espinosa-Gongora, Poul Bækbo

**Affiliations:** 1Department of Animal and Veterinary Science, Aarhus University, Blichers Allé 20, 8830 Tjele, Denmark; 2Triolab AS, Vallensbækvej 35, 2605 Brøndby, Denmark; manie@triolab.dk; 3SEGES Innovation P/S, Agro Food Park 15, 8200 Aarhus, Denmark; mefr@seges.dk (M.E.F.); macbirpo@fibermail.dk (P.B.); 4National Food Institute, Technical University of Denmark, Henrik Dams Allé, Building 204, 2800 Kongens Lyngby, Denmark

**Keywords:** LA-MRSA, CC398, pig production, occupational hazard

## Abstract

Livestock-associated methicillin-resistant *Staphylococcus aureus* (LA-MRSA) in pig production presents an occupational hazard for farm workers. Understanding the levels of LA-MRSA is important to assess worker exposure. In the present study, we monitored pigs from 30 kg until slaughter and found that the nasal carriage of LA-MRSA, measured as the number of bacteria per sample, gradually decreased during the rearing period, although all samples remained positive throughout. LA-MRSA levels in the air were generally low to moderate, but dust samples contained high levels, reaching up to 37,272 bacteria per gram of dust.

## 1. Introduction

Methicillin-resistant *Staphylococcus aureus* of the type CC398, also known as livestock-associated MRSA, LA-MRSA, first appeared in France and the Netherlands around 2004 [1] and has since spread across most countries in Europe and beyond [2,3]. LA-MRSA is particularly associated with pigs but has also been found in a range of other animal species, including poultry, mink, horses, dogs, and cats [4,5,6,7,8,9,10,11]. Most animals are healthy carriers of LA-MRSA CC398, but it may also cause infections, such as skin, urinary tract, and wound and joint infections in pigs; abscesses in mink; mastitis in dairy cattle; or skin-, ear-, wound-, or urinary tract infections in dogs [12,13,14,15].

In Denmark, LA-MRSA is now widespread [16,17]—a dramatic development since the first survey in 2008 [3]. Transmission to humans occurs mainly through direct or indirect contact with infected animals, whereas food is not considered a major source [18,19]. Pig herd workers will invariably be exposed and frequently carry LA-MRSA, either transiently or permanently [20,21,22]. A German study [23] showed an increased risk of LA-MRSA positivity in household members of pig and poultry herd workers and among those who regularly visit herds. Very high carrier rates among farmers, livestock veterinarians, and slaughterhouse personnel have been confirmed [24] and further detailed by Goerge et al. [25].

Likewise, Cuni et al. [21] found that a very high percentage of individuals in contact with livestock were infected, while 4–5% of their household members were carriers. Infection beyond these groups is less common, but nevertheless, a part of the infections with MRSA in humans are attributed to LA-MRSA, especially in areas with high livestock density [21], and this number of infections has also been increasing in Denmark but has now reached a plateau [16,26], likely because the number of infected pig herds is no longer increasing. While few people die from LA-MRSA infections, the bacterium generally causes similar infections to other MRSA lineages [21].

Recent studies have shown that short-term visitors to infected herds can become transiently contaminated but typically clear the bacterium within hours or days [27]. There was a relationship between LA-MRSA levels in air samples and nasal swab samples from healthy volunteers [27]. Bos et al. [28] found that both duration of exposure and levels of LA-MRSA in air had an impact on nasal carriage among farm workers.

Reducing LA-MRSA levels in barn air will decrease the discharge of bacteria into the environment through the ventilation system. LA-MRSA can be detected in both air samples and soil surfaces near infected pig herds [11,29], and it was recently demonstrated that LA-MRSA can be found in pig manure and in soil samples from fields where pig manure has been spread [30]. It is not yet known whether and to what extent this discharge contributes to the spread of livestock MRSA to humans in the local community, but although the risk is considered to be low, there is evidence that it may play a certain role [31]. The level of LA-MRSA that poses a negligible risk to employees and visitors carrying the infection out of the stables remains unknown.

Understanding LA-MRSA levels in pig herds across different age groups and sections is therefore relevant. However, few studies so far have quantified LA-MRSA; most merely indicate the presence or absence of LA-MRSA [32,33,34,35]. Previous investigations have suggested that the highest LA-MRSA levels are in the weaning sections and the lowest in the gestation sections [14], and some studies report a reduction in the proportion of positive samples from the beginning to the end of the fattening period [32,33,34]. The proportion of positive samples has been shown to depend on many other factors not associated with the pigs, e.g., bedding, cleaning and disinfection routines, general hygiene, and there can be considerable herd-to-herd differences [32,33,34,36,37].

Farm workers are exposed to LA-MRSA through direct contact with animals or contact with contaminated environmental sources, such as dust, or inhaling contaminated air. Therefore, we have focused on these sources. The aim of this investigation was to quantify LA-MRSA levels in a herd of slaughter pigs and track any changes over time. LA-MRSA levels in pigs were monitored from 30 kg until slaughter by analyzing nasal swabs collected from the pigs at three time points: arrival from the weaning section, midway through the rearing period, and before slaughter. Additionally, LA-MRSA levels were measured in air and dust samples within the herd.

## 2. Materials and Methods

### 2.1. Herd

The study was conducted on a conventional production herd with a capacity of 4000 slaughter pigs. It was located in a rural area with approx. 3 km distance to the nearest village and approx. 10 km to the nearest town. The region is a typical agricultural area with a high density of both pigs and cattle. The herd received 11-week-old pigs weighing approximately 30 kg from a nursery herd located at another site owned by the same producer. Pigs were crossbreeds (Landrace × Yorkshire × Duroc) and were part of an SPF (specific pathogen-free) herd, reinfected with *Mycoplasma hyopneumoniae* and *Actinobacillus pleuropneumoniae* serotypes 6 and 12.

The facility was organized into eight identical sections located as four coherent buildings, two on each side of a central aisle, with each building holding two identical sections (Figure 1). Each section was 493 m^2^ and 1380 m^3^ and designed to house 500 slaughter pigs distributed in 28 pens, each 2.4 m × 6.3 m. The 28 pens were arranged in two rows of 14 pens, separated by a central aisle. The ventilation was a common diffuse ventilation system, and the pigs were fed a standard wet feed. All 500 pigs in a section shared the same airspace, which was separated from the airspace of the other sections. The eight sections were stocked with pigs over a period of 4–5 weeks, with two sections being filled at a time, and a strict all-in, all-out protocol was followed. The pigs stayed in the herd for approx. 11 weeks, with the first pigs from a section being sent to slaughter after approx. nine weeks. Thereafter, the whole section was cleaned and washed with a high-pressure washer, but no detergent or disinfectant was used. After washing, the house was dried at 30 °C for two days using a heat cannon. The entire time cycle of each section was thereby 13 weeks—filling, rearing, emptying/slaughter, cleaning, and drying period, after which a new batch of pigs was introduced.

The herd was part of a larger study testing the potential effect of four different intervention strategies to reduce LA-MRSA under commercial rearing conditions without altering production speed or scaling down due to the experiments and using the full production facility and not just a small experimental facility. The interventions were: (1) recirculation of air through a device that washed away dust particles and treated the air with ozone and UV light; (2) daily spraying of the premises and animals with electrolyzed water, along with adding it to the feed; (3) spraying the premises and the pigs with a dust-precipitating chemical twice a day; and (4) application of a commercial disinfectant powder product for livestock herds. None of the technologies showed any effect on LA-MRSA levels, feed conversion ratios, daily weight gain, or the prevalence of common pig pathogens such as enterotoxigenic *Escherichia coli*, *Brachyspira pilosicoli,* or *Lawsonia intracellularis* (see Bækbo et al. [38] for details). For each intervention test, a test group and a control group were run in parallel, one group = one section. Three of these strategies were tested twice, i.e., in two sections, and the last one only once. In this investigation, data from all control groups in those experiments were analyzed, which included a total of seven batches of pigs, representing 3500 animals. Although no differences between test groups and control groups were found [38], data from the test groups were not included since these pigs had been exposed to a treatment.

### 2.2. Sampling

Samples were collected from seven batches of 500 pigs as described below. The batches were not run in parallel but were initiated and sampled successively over 35 weeks, as shown in Appendix A.

#### 2.2.1. Nasal Swab Samples from Pigs

Nasal swab samples were collected using flocked swabs (ESwab, Copan Diagnostics Inc., Murrieta, CA, USA). The swabs were rotated three times in each nostril, approx. 1 cm inside, and then placed in its corresponding tube containing 1 mL of Amies Transport Medium [15]. For each batch, 24 randomly selected pigs were sampled at three time points: (i) the first week after arrival, (ii) during the fourth week, and (iii) during the eighth week, just before the first pigs were sent for slaughter.

#### 2.2.2. Dust Samples

Dust was sampled by brushing it from horizontal surfaces of equipment into sterile plastic containers with a sterile gloved hand. Two dust samples were taken at each sampling, representing different ends of the section. Dust samples were collected at the same time as the nasal swabs. In most cases, it was not feasible to sample dust at the first sampling in week 1, due to insufficient dust accumulation.

#### 2.2.3. Air Samples

Air samples were collected using a Sartorius Airport MD8 device (Sartorius AG, Göttingen, Germany). Two samples were taken on each sampling occasion, resulting in a total of 42 air samples. Samples were collected 1.5 m above the floor to simulate the typical height at which farm workers are exposed to airborne substances. Samples were collected over a 5 min period as the person walked slowly down the aisle to ensure a representative sample from the entire section, i.e., both samples were collected at the same place but covering the entire aisle. Air was filtered through a gelatin filter, capturing all bacteria. The Sartorius machine was decontaminated with ethanol before each sampling, and the filter was handled with sterile tweezers. Air samples were collected at the same time as the nasal swab samples.

#### 2.2.4. Transport from Herd to Laboratory

Samples were placed in an insulated box with cooling elements immediately after collection. The box was sent the same afternoon to the laboratory by courier transport and was received the following morning at the laboratory. Analyses began immediately upon arrival.

### 2.3. Culture and Identification

#### 2.3.1. Nasal Swab Samples

MRSA in swabs was quantified by directly inoculating 100 µL of the transport medium, along with a 10^−1^ dilution in phosphate-buffered saline onto selective Brilliance MRSA2 agar plates (Oxoid, Basingstoke, UK). The plates were then incubated at 37 °C for 18–24 h. Colonies were counted and expressed as colony-forming units (cfu)/swab. In case of an overgrowth of MRSA colonies, an additional 10-fold dilution was prepared and plated.

Samples negative by direct inoculation were further examined by enriching the remaining transport medium in Mueller–Hinton broth with 6.5% NaCl, followed by incubation at 37 °C for 18–24 h to confirm the presence of MRSA. Subsequently, 10 μL of the enriched broth was streaked onto Brilliance MRSA2 agar and read after incubation at 37 °C for 18–24 h.

Representative MRSA suspect colonies were subcultured on blood agar plates and verified as *Staphylococcus aureus* by MALDI-TOF as previously described [39]. Verification of MRSA by PCR detection of the *mecA* gene was performed on individual colonies as described by Agersø et al. [17].

#### 2.3.2. Dust Samples

Dust was processed by suspending 0.5 g in 4.5 mL sterile saline, followed by serial 10-fold dilutions (10^−1^, 10^−2^, and 10^−3^) in sterile saline. For each dilution, 100 µL was plated onto Brilliance MRSA2 agar plates and incubated at 37 °C for 18–24 h. Colonies were counted and calculated as cfu/g of dust. Representative colonies were tested to confirm MRSA as described above.

#### 2.3.3. Air Samples

Immediately after sampling at the farm, gelatin filters were transferred from the Sartorius air sampler to MRSA2 agar plates. Upon arrival at the laboratory, the plates were incubated at 37 °C for 18–24 h. After incubation, colonies were counted and verified as described above. Counts were calculated and expressed as cfu/m^3^ air.

### 2.4. Statistics

Colony-forming units were log-transformed and tested for normal distribution using the Shapiro–Wilk test. A one-way ANOVA was used to compare counts from the first, second, and third sampling times, followed by pairwise comparisons conducted using a *t*-test and a Tukey’s HSD multiple comparison test. If the data were not normally distributed according to the Shapiro–Wilk test, a Mann–Whitney U test was used instead. Similar comparisons were made between the seven groups at each sampling time. Calculations and visualizations were performed using Statistics Kingdom online calculators (www.statskingdom.com accessed 7 November 2024). For nasal swabs testing negative by direct inoculation but positive after enrichment, counts were assigned half the detection limit (5 cfu), while negative samples were assigned 1 cfu, which converts to 0 after log-transformation.

## 3. Results

The mean counts across batches and sample types are shown in Table 1, with batch differences described in more detail below.

### 3.1. Nasal Swab Samples

From each of seven batches of 500 pigs, 24 nasal swab samples were collected at three time points during the rearing period, i.e., 168 samples per batch, totaling 504 samples. All nasal swab samples were found positive for LA-MRSA, 494 of them by direct plating, 10 only after enrichment. The LA-MRSA count in a sample was 353,000 cfu, whereas the 10 samples that were positive only after enrichment presumably had 1–10 cfu. Two samples had more than 100,000 cfu, while 22 had between 10,000 and 100,000, all of them at the first or second sampling.

There were statistically significant differences both between batches of pigs at the same sampling time and between sampling times. Figure 2 shows the counts for each batch at the three time points. It is evident that even though the pigs were treated identically from birth to slaughter, there was significant batch-to-batch variation (1. sampling: *p* = 2.471 × 10^−10^, 2. sampling: *p* = 1.334 × 10^−5^, 3. sampling: *p* = 6.443 × 10^−4^).

When LA-MRSA levels across all samples were compared at each sampling time (Figure 3), a significant reduction was observed from the first to the second sampling (*p* = 6.66 × 10^−16^) and again from the second to the third sampling (*p* = 3.668 × 10^−9^). The mean count at first sampling was 1387 cfu/swab (log_10_ cfu = 3.142), which decreased by 81.5% at the second sampling and by 94.1% at the third.

For all batches except for batch 7, there was a decrease in counts from first to second sampling and for all batches again from second to third sampling. In general, the largest reduction in counts occurred between the first and second sampling.

### 3.2. Dust Samples

At the first sampling, the section had recently been cleaned, yielding little or no dust in most cases. Therefore, only counts from the 28 samples collected during the second and third sampling are shown (Figure 4). There was no significant difference in log_10_ cfu/g between the two time points (*p* = 0.8541). The highest count in a sample was 37,272 cfu/g, and the overall mean was 17,185 cfu/g. One sample was positive only after enrichment, suggesting 1–10 cfu/g. However, the second sample taken at the same time from that batch had 5454 cfu/g. All other samples had >1000 cfu/g.

### 3.3. Air Samples

Counts of LA-MRSA in the 42 air samples were low, with a mean of 63 cfu/m^3^, a median of 28 cfu/m^3^, and a range of 0–568 cfu/m^3^. Since there was no statistical difference between counts at the three sampling points, all counts are shown together in Figure 5.

## 4. Discussion

### 4.1. LA-MRSA Quantification

Most LA-MRSA studies in pig herds have focused on presence/absence rather than quantification. In this study, a quantitative approach revealed a 94% reduction in average LA-MRSA counts in nasal swabs from 30 kg pigs until slaughter, a finding that qualitative methods would not have captured. Quantitative data are therefore essential for identifying high-exposure times and locations, as well as for discussing potential intervention points. This approach also broadens the possibilities for evaluating intervention effectiveness and designing future studies to reduce LA-MRSA exposure for farmers, making them more accurate by quantifying LA-MRSA loads.

This investigation shows that consistent LA-MRSA levels across batches cannot necessarily be expected, even when pigs originate from the same breeding farm and are raised under similar conditions, such as those present at this farm. The observed variation suggests that other factors, unrelated to the pigs, significantly influence LA-MRSA levels, highlighting the importance of optimizing the amount of quantitative data to manage LA-MRSA exposure and understand its dynamics within farms. All batches of pigs in the present study were raised under the same conditions concerning temperature, management, feeding, and other factors. Therefore, this batch-to-batch variation was surprising, and there is currently no explanation for it.

Hansen [14] found the highest LA-MRSA levels in nasal swabs from weaning sections, followed by farrowing sections, with the lowest levels found in gestation sections (no slaughter pigs were included in that study). Our study observed higher LA-MRSA nasal counts than those in Hansen’s findings but lower levels in air samples [14]. Both studies found a correlation between air and nasal counts. Verkola et al. [40] reported even lower LA-MRSA levels in nasal swabs, indicating farm-specific dynamics likely influenced by multiple factors. The marked differences in LA-MRSA levels between individual pigs raise the possibility that some individual pigs act as super-shedders of LA-MRSA, as previously suggested (persistently colonized pigs carrying high LA-MRSA levels) [41]. However, because we did not collect samples from the exact same animals on the three sampling occasions, we cannot confirm the presence of such pigs. Altogether, the observed variations highlight the need for farm-specific LA-MRSA reference counts when designing effective control measures, and studies focused on reducing LA-MRSA exposure.

Despite the reduction of LA-MRSA levels in pigs towards the end of the production cycle, we found high LA-MRSA levels in dust samples during the second and third samplings, suggesting that LA-MRSA can survive in dust for extended periods. The survival of LA-MRSA on various surfaces is well documented, including on polypropylene (half-life 11–16 days) and stainless steel (half-life 2–8 days), while survival was shorter on other surfaces such as concrete [42]. These findings emphasize the importance of cleaning and disinfection and of considering the materials present in the stables when attempting to reduce LA-MRSA. Surface materials, bedding, and cleaning and disinfection procedures all play significant roles in the persistence of LA-MRSA throughout the production cycle. For example, pigs on straw bedding were more often negative for LA-MRSA than those on other types of flooring, possibly due to competitive interactions between LA-MRSA and other microorganisms from the straw [33]. Competitive associations between LA-MRSA and other bacteria colonizing the nose and skin of pigs have also been reported [43]. Similar to bedding, cleaning without disinfection was associated with more LA-MRSA-free animals at the end of the rearing period compared to cleaning combined with disinfection [33]. This may seem contradictory; however, some researchers have suggested that the use of quaternary ammonium-based disinfectants may be counterproductive due to the possible presence of genes conferring resistance to these compounds [44].

LA-MRSA air contamination affects not only indoor environments but also impacts areas outside farm stables. The large volume of air exhausted from pig farms—several thousand m^3^ per hour, especially during warm periods—spreads LA-MRSA widely with the wind. Schultz et al. [29] demonstrated this by detecting LA-MRSA in soil samples collected in the vicinity of pig farms, particularly downwind. In addition, LA-MRSA has been found in pig manure, where it may survive for extended periods of time and be spread on agricultural fields when the manure is used as fertilizer [30]. Consequently, LA-MRSA presents a significant true one-health hazard, impacting pigs, farm workers, the surrounding community, and the environment.

National recommendations to reduce human exposure to LA-MRSA from pigs have been issued, which include changing clothes and showering when entering and exiting farms [18]. Studies have shown that wearing masks reduces human exposure and the nasal load of LA-MRSA [27]. This study may add that while exposure from direct contact with the pigs decreases significantly through the rearing period, loads of LA-MRSA remain high in dust, so farm workers should probably avoid contact with dust as much as possible.

We did not seek permission to collect samples from farm workers in this study, but a previous one found levels in the range 0–650 cfu/sample and showed a certain correlation between air and nasal loads in healthy volunteers [27]. Most investigations on humans, however, have only tested for the presence or absence of MRSA, leaving limited data on actual loads in farm workers. These observations suggest some degree of dose-response relationship between exposure and human contamination. Colonized persons are at higher risk of infection [25]. Therefore, reducing LA-MRSA amounts in the stable air and dust could reduce exposure and bacterial loads carried by farm workers as they leave the stables, potentially shortening the duration they remain positive for LA-MRSA after leaving the barn.

Our study provides abundant data produced with selected methods for sample collection and laboratory analysis, which were piloted prior to the large intervention trials mentioned above. These methods were easy to implement on farms, allowed for the collection of large sample quantities in a single campaign, and were straightforward to continue with subsequent laboratory analyses. Though it is more labor-intensive to make ten-fold dilutions and count colonies than to just register growth/no growth, the results are considerably more informative.

### 4.2. Reducing LA-MRSA Levels and Exposure

The factors influencing the success or failure of interventions aimed at reducing LA-MRSA in pig farms are only marginally understood. While commonly used disinfectants have shown efficacy against MRSA in vitro, their effectiveness in real-life conditions at production farms remains uncertain. Several studies indicate that cleaning and disinfection between batches can effectively reduce LA-MRSA levels in the subsequent batch [32,33,34,37]. Washing and disinfecting the skin of sows before and after farrowing temporarily reduced LA-MRSA levels [36]. Other tested interventions, such as the use of biocides, ozone, UV light, and dust reduction, have also shown no significant effect compared to control groups [38].

Experience suggests that once LA-MRSA is introduced into a pig farm, it will remain positive. This is reflected in the lack of reports describing spontaneous eradication of LA-MRSA in pig farms and is supported by simulation studies suggesting that reducing MRSA levels is challenging [45,46]. It remains unclear whether the main reservoir within the farm is the environment or the animals. Some studies suggest the presence of persistently colonized pigs [41], while others note that pigs in organic outdoor production systems may remain LA-MRSA-free [32].

The only successful intervention reported to date is Norway’s zero-tolerance strategy, in which LA-MRSA-infected herds are immediately culled, cleaned, and disinfected, with contact herds also being tested and culled if found positive [47]. This strategy is costly and likely sustainable only in countries with low LA-MRSA prevalence, such as Norway. In countries with high density of pig farms and frequent pig transport, such as Germany, Denmark, or Belgium, this approach may be impractical due to the lack of economic incentives and the non-food-borne nature of LA-MRSA.

Efforts to reduce antibiotic use have shown mixed results in LA-MRSA prevalence. Dutch studies indicated that reducing antibiotic use correlated with lower MRSA prevalence in pigs and cattle [48,49]. However, other studies found no such correlation [50]. A Danish study showed that reducing zinc and tetracycline reduced LA-MRSA in pigs, but this was under experimental conditions and not in production settings [51]. We did not perform antibiotic susceptibility tests on isolates from this study, but previous studies indicate that LA-MRSA CC398 is resistant to tetracyclines and frequently resistant to spectinomycin, streptomycin, and macrolides, which are commonly used in pigs. In contrast, resistance to sulfonamides with trimethoprim and florfenicol is low [14]. Antibiotic use may contribute to the selection and persistence of LA-MRSA CC398 on farms.

This study focused on LA-MRSA in nasal swab samples, air, and dust, since they are considered major sources of exposure for farm workers. A limitation of the study was that no samples were collected from farm workers, so it would be valuable to investigate correlations between loads in humans, pigs, and environmental sources. Likewise, any management, environmental, or other factors that might affect loads of LA-MRSA should be investigated.

## 5. Conclusions

This study demonstrated an approx. 95% reduction in LA-MRSA counts in nasal swab samples in pigs from weaning to slaughter. All nasal swab samples were positive for LA-MRSA, which highlights the importance of quantification of the bacterium for making estimations of changes in carriage. Significant batch-to-batch differences in nasal carriage of LA-MRSA were found, which demonstrates that although all pigs originated from the same herd and were raised under identical conditions, there are other factors that influence the carriage of LA-MRSA. Levels of LA-MRSA in air samples were in general low, mean counts 63 cfu/m^3^ (range 0–568), but high in dust samples, mean 17,186 cfu/g. Dust may therefore be a significant source of exposure for herd workers. More research on quantification of LA-MRSA in pig farms across farm size, animal age groups, management, and breeding procedures is essential to design and validate effective intervention strategies.

## Figures and Tables

**Figure 1 vetsci-11-00558-f001:**
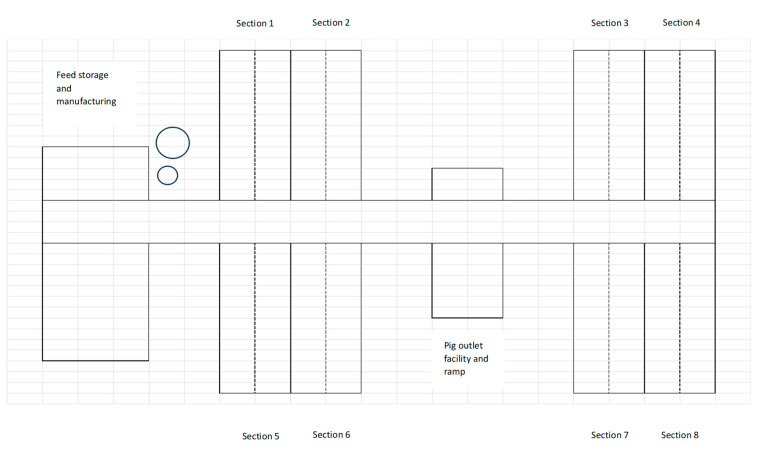
Structure of the experimental facility. The herd consisted of four identical modules, each of which were split into two sections with an aisle in the middle. Each section had space for 500 pigs.

**Figure 2 vetsci-11-00558-f002:**
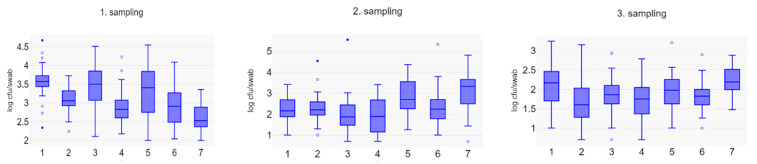
Box plot of log_10_ cfu of LA-MRSA in nasal swab samples of 24 pigs per batch in seven batches sampled on three occasions. Numbers along the x-axis refer to batch numbers.

**Figure 3 vetsci-11-00558-f003:**
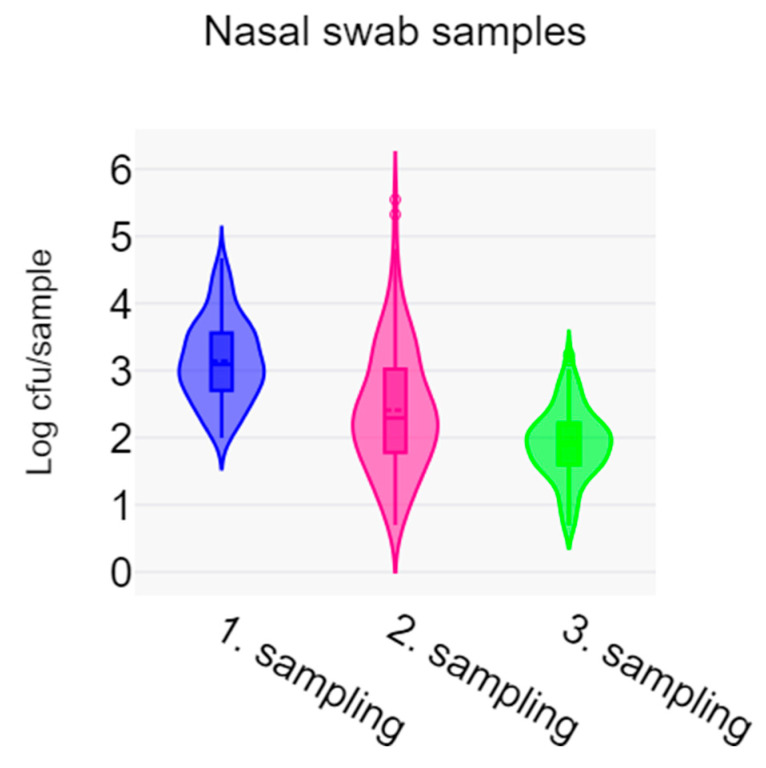
Violin plot of log_10_ cfu of LA-MRSA per swab for all samples at the three time points.

**Figure 4 vetsci-11-00558-f004:**
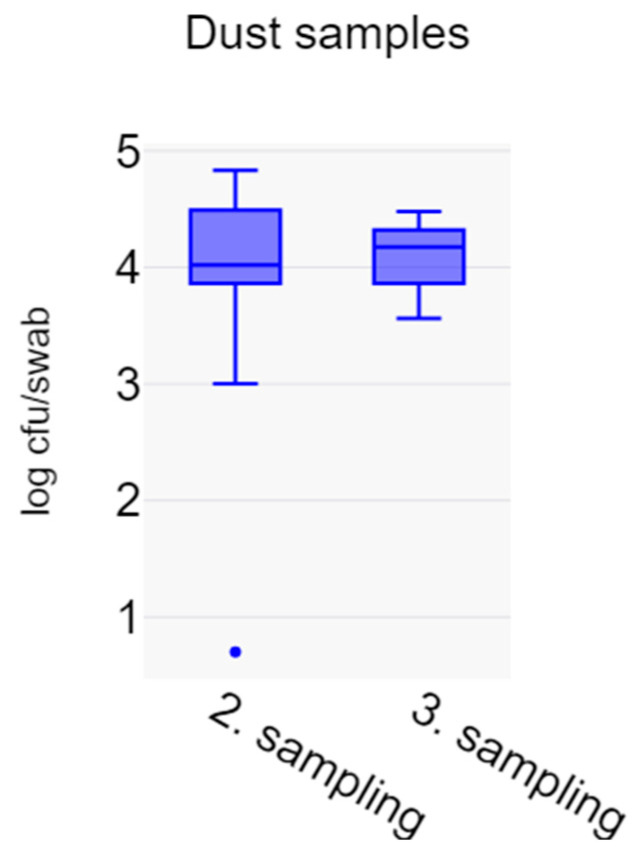
Box plot of LA-MRSA counts in dust samples, expressed as log_10_ cfu/g dust.

**Figure 5 vetsci-11-00558-f005:**
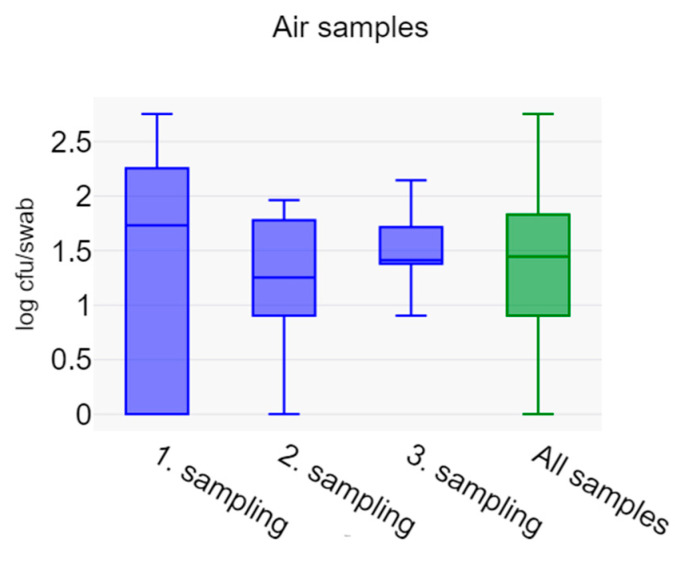
Box plot of LA-MRSA counts in air samples, expressed as cfu/m^3^. The graph includes samples at each sampling point and the total for all 42 air samples that were collected from the three samplings.

**Table 1 vetsci-11-00558-t001:** Summary of LA-MRSA loads in all samples across sampling points and sample types.

	Sampling 1 Counts ± SD	Sampling 2 Counts ± SD	Sampling 3 Counts ± SD
Nasal swabs (Log_10_ cfu/sample)	3.14 ± 0.61 ^a^	2.41 ± 0.92 ^b^	1.91 ± 0.53 ^c^
Dust (Log_10_ cfu/g)	ND	3.88 ± 1.03	4.09 ± 0.30
Air (Log_10_ cfu/m^3^)	1.41 ± 1.03	1.14 ± 0.71	1.48 ± 0.36

Figures within rows with different superscript letters, ^a^, ^b^, or ^c^, differ statistically (*p* < 0.05).

## Data Availability

The datasets generated and analyzed during this study are available from the authors upon reasonable request.

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
