# Peer review of "Development of Livestock-Associated Methicillin-Resistant *Staphylococcus aureus* (LA-MRSA) Loads in Pigs and Pig Stables During the Fattening Period"

_vetsci, 2024, doi:10.3390/vetsci11110558_

Round 1

Reviewer 1 Report

Comments and Suggestions for Authors

Comments on the Quality of English Language

 Minor editing of English language required.

Author Response

General Comments

  • Language: The manuscript employs clear and professional language, which is suitable for an

academic audience.

Thank you

  • Implications for Practice: It is essential to emphasize how farm management can integrate the findings, particularly by focusing on practical strategies to minimize LA-MRSA exposure.

There are already phrases in the Discussion section, which address this in general terms, but now we have inserted a few lines at the end of section 4.1. that applies specifically to this study.

  • Tables: Including a table that summarizes LA-MRSA levels across different sampling points and sample types would enhance clarity and improve reader comprehension and engagement.

We have now added a small table, Table 1, at the beginning of the Results section, which shows averages of counts and indicating figures that are statistically significant. This has then made Supplementary Figure 1 obsolete, since it is redundant information, so we have removed that.

Specific Comments

  • Statistics

Lines 204-205: While using a t-test for pairwise comparisons following a one-way ANOVA is

common, it is advisable to use dedicated post hoc tests for more robust conclusions.

  1. Multiple Comparisons Problem: Conducting multiple t-tests after an ANOVA increases the risk of Type I error (false positives). To mitigate this risk, consider applying a correction method, such as the Bonferroni correction or Tukey's HSD (Honestly Significant Difference) test, which is specifically designed for pairwise comparisons following ANOVA.
  2. Post Hoc Tests: Many researchers advocate for the use of post hoc tests (e.g., Tukey's HSD, Scheffé's test, or Dunnett's test) that are tailored for multiple comparisons after an ANOVA.

These tests effectively address the multiple comparisons issue and yield more reliable results

Thank you for this comment. In fact, the programme that we used, automatically performs the Tukey’s HSD test, so we have these results already in the test report and they are the same. We have included this information now in the M&M section. Not sure whether it is needed to add anything in the Results since nothing is changed.

Reviewer 2 Report

Comments and Suggestions for Authors

Livestock-associated methicillin-resistant Staphylococcus aureus (LA-MRSA) in livestock production is an occupational hazard. The manuscript investigated development of LA-MRSA loads in pig from 30 kg until slaughter. The study is very straightforward. The authors quantified LA-MRSA levels in Nasal Swab Samples of pigs, Dust Samples and Air Samples.

As, LA-MRSA is a significant public health concern due to its zoonotic potential, I request the authors to quantify LA-MRSA levels in individuals in close contact with pigs.

The readers will also be curious to know about the antimicrobial resistance status of MRSA isolates. The authors may look into that if it is within the scope of the study.

 Moreover, the introduction looks very lengthy. Kindly make it shorter keeping the major points intact.

Author Response

Livestock-associated methicillin-resistant Staphylococcus aureus (LA-MRSA) in livestock production is an occupational hazard. The manuscript investigated development of LA-MRSA loads in pig from 30 kg until slaughter. The study is very straightforward. The authors quantified LA-MRSA levels in Nasal Swab Samples of pigs, Dust Samples and Air Samples.

As, LA-MRSA is a significant public health concern due to its zoonotic potential, I request the authors to quantify LA-MRSA levels in individuals in close contact with pigs.

We did not have any ethical permissions to collect samples from farm workers, so this information we do not have. We have instead included some information in the Discussion at the end of section 4.1 about findings from another study. This was of course another farm, which may have had different loads of LA-MRSA in air and dust but still better than nothing.

The readers will also be curious to know about the antimicrobial resistance status of MRSA isolates. The authors may look into that if it is within the scope of the study.

Similarly, we did not perform any MIC determinations on isolates from the farm. What we do know, though, is the resistance status of LA-MRSA on a national level. We have inserted a few phrases about this in the Discussion at the end of section 4.2. We have no reason to believe that the situation on this farm should be different.

 Moreover, the introduction looks very lengthy. Kindly make it shorter keeping the major points intact.

We are not sure that we agree to this, but now at least we have axed some lines and rephrased and moved some lines to the Discussion in connection with the text mentioned above concerning human carriage.

Reviewer 3 Report

Comments and Suggestions for Authors

the authors use nasal swab in several places during the manuscript. Especially in the introduction it needs to be clear these are from the pigs and not people. it becomes clear later in the manuscript but is unclear in the introduction. 

the sentence in line 74 to 79 is unclear. I would suggest simplifying the sentence into two or three sentences to improve the meaning.

Comments on the Quality of English Language

No problems with the English.

Author Response

the authors use nasal swab in several places during the manuscript. Especially in the introduction it needs to be clear these are from the pigs and not people. it becomes clear later in the manuscript but is unclear in the introduction.

Thank you for this comment – we were not aware of this potential confusion. Now we have stated this in both the Abstract and the Introduction sections.

the sentence in line 74 to 79 is unclear. I would suggest simplifying the sentence into two or three sentences to improve the meaning.

Thank you for pointing this out. We have split the sentence and rephrased it a bit and moved it to the Discussion.

Reviewer 4 Report

Comments and Suggestions for Authors

In this study, the authors monitored the LA-MRSA levels in nasal samples of pigs from 30 kg until slaughter and air and dust samples from pig stables. They pointed out that the number of LA-MRSA per sample decreased gradually during the rearing period, although all samples remained positive throughout. LA-MRSA levels in air samples were generally low to moderate, but dust samples contained high numbers of LA-MRSA.

 Strength:

These findings indicate that the high numbers of LA-MRSA in dust may pose an occupational hazard to farm workers.

 Points

1. The background information about the experimental pig herd is not well described. The location of the pig farm? The surroundings? Distance from residential areas, etc.?

 2. Lines 124-125, “The time cycle of a section was thereby 13 weeks, after which a new batch of pigs was introduced.” It is not clear here. Is the time cycle for the whole section or a particular section?

 3. Lines 129-130, “The herd was used in a big trial where the potential effect of four different intervention strategies against LA-MRSA was tested under full-scale production conditions. What is “full-scale production conditions”?

 4. How to find reference 40?

 5. Lines 139-141, “In the present investigation, we analyzed data from all control groups in those experiments, which included a total of seven batches of pigs, representing 3,500 animals.” What is the design of the big trial described by the author? How many groups are there? How many control groups are there? What are they? Why seven batches?

 6. The author described the sampling time as "the first week, the fourth week, etc.". For seven batches of pigs, are the three sampling times on the same day within the week or are there differences?

 6. Lines 155-156, “Two dust samples were taken at each sampling, representing different areas of the section.” On what basis were the areas selected here?

 7. For air sampling, what is the basis for selecting where to sample in the pig house?

 8. Supplementary Figure 1 should have bars.

 9. It is recommended that the test results of 28 dust samples and 42 air samples be presented in the supplementary materials.

Author Response

In this study, the authors monitored the LA-MRSA levels in nasal samples of pigs from 30 kg until slaughter and air and dust samples from pig stables. They pointed out that the number of LA-MRSA per sample decreased gradually during the rearing period, although all samples remained positive throughout. LA-MRSA levels in air samples were generally low to moderate, but dust samples contained high numbers of LA-MRSA.

 Strength:

These findings indicate that the high numbers of LA-MRSA in dust may pose an occupational hazard to farm workers.

Yes, and we have now added a couple of lines in the Discussion section to make this more clear.

 Points

  1. The background information about the experimental pig herd is not well described. The location of the pig farm? The surroundings? Distance from residential areas, etc.?

We have added some more information about the farm. However, the contract between the farm owner and the research team is confidential, so we are not allowed to point out its position on a map or reveal herd registration number.

  1. Lines 124-125, “The time cycle of a section was thereby 13 weeks, after which a new batch of pigs was introduced.” It is not clear here. Is the time cycle for the whole section or a particular section?

Each batch took up a time slot of 13 weeks, which included filling, rearing, emptying/slaughter, cleaning, and drying. The different sections were not parallel, so it was more so that in one week, section 1 and 2 were filled, the following week section 3 and 4, and so on. This also meant that sampling was carried out continuously, since week 1 for section 1 and 2 was not calendar-wise the same as week 1 in section 3 and 4, etc. This way, samples were collected on the farm almost weekly from week 39 one year to week 21 the following year. We have tried to formulate this better in the M&M.

  1. Lines 129-130, “The herd was used in a big trial where the potential effect of four different intervention strategies against LA-MRSA was tested under full-scale production conditions.” What is “full-scale production conditions”?

We simply mean that it was carried out on a farm running under commercial condition without scaling or slowing down due to the experiments. Also, it was the whole farm that was included in the project, and not a small experimental facility, which does not necessarily reflect real life conditions. We have pointed this out better now in the M&M section.

  1. How to find reference 40?

Ah, right, we understand, sorry. We have now added a link in the reference list, so it should be possible to get access directly – open access and free to use, provided proper reference is made. It is still in Danish, though, but Google Translate can sometime make miracles.

  1. Lines 139-141, “In the present investigation, we analyzed data from all control groups in those experiments, which included a total of seven batches of pigs, representing 3,500 animals.” What is the design of the big trial described by the author? How many groups are there? How many control groups are there? What are they? Why seven batches?

We performed a large intervention study, testing four different intervention strategies. Three of these strategies were tested twice, i.e., in two sections, and the last only once. Such a study is quite laborious and expensive, so seven batches was what the involved companies were willing to invest. So, this was mainly for economic reasons, and since we did not see any effect on LA-MRSA in the experimental groups compared to the controls, in any of these seven batches, we decided to stop. There were some power calculations behind the choice of numbers of samples, etc., to be able to document an effect of the intervention, but they are not really relevant to go into here. For each intervention, an experimental section and a control section were run in parallel. However, since the experimental groups had been exposed to the intervention products, we did not want to use those for the present paper, but all the control sections (one section = one group) were raised completely as the farmer used to do, so we could use the counts from these sections in the present paper.

We have tried to expand a bit on line 139-141. This together with the sampling plan – see below - should make everything clearer (we hope).

  1. The author described the sampling time as "the first week, the fourth week, etc.". For seven batches of pigs, are the three sampling times on the same day within the week or are there differences?

No, the sections were not tested at the same time. Each of the intervention studies were initiated successively, since they were very labour intensive, so the control groups were also tested successively. The sampling plan can be shown like this:

Week no.

Intervention strategy/batch

39

40

41

42

43

44

45

46

47

48

49

50

51

52

1

2

3

4

5

6

7

8

9

10

11

12

13

14

15

16

17

18

19

20

21

1/1

X

X

X

2/2

X

X

X

3/3

X

X

X

1/4

X

X

X

2/5

X

X

X

3/6

X

X

X

4/7

X

X

X

We have now included the sampling plan as a Supplementary Table 1. Maybe it will give better clarity over both sampling and time span, and maybe also clarify some of the other points raised above.

  1. Lines 155-156, “Two dust samples were taken at each sampling, representing different areas of the section.” On what basis were the areas selected here?

Well, randomly chosen, horizontal surfaces in different ends of each section. We anticipated a fairly even distribution of LA-MRSA in each section due to air circulation and, in fact, they were. This was reflected by the counts in the two samples, which were in most cases more or less the same. So, as it turned out, it was not important where the dust was collected.

We have slightly modified this section in the M&M

  1. For air sampling, what is the basis for selecting where to sample in the pig house?

The aisle was the only feasible place to sample. Otherwise, we would have to crawl from pen to pen carrying the equipment, and that was not really manageable. So, the two samples were in fact taken in the same place but collected at the entire length of the aisle. This piece of information has now been added to the M&M section.

  1. Supplementary Figure 1 should have bars.

We have included a new Table 1 on request from one of the other reviewers summarizing the results across batches and sample types. Therefore, we have decided to omit Supplementary Figure 1.

  1. It is recommended that the test results of 28 dust samples and 42 air samples be presented in the supplementary materials.

We have not included a supplementary file containing the raw data. All raw data are collected in an Excel file, however including names and other information that cannot be made publicly available due to GDPR regulations. We have added a Data Availability Statement saying that the dataset can be obtained from us upon reasonable request. It will need some anonymization from our side first, though.  

Reviewer 5 Report

Comments and Suggestions for Authors

The study entitled “Development of LA-MRSA Loads in Pigs and Pig Stables during the Fattening Period investigates the development of livestock-associated methicillin-resistant Staphylococcus aureus (LA-MRSA) in pigs and their environment during the fattening period. Through the quantification of LA-MRSA in nasal swabs, air, and dust, the study reveals a significant reduction in nasal carriage over time, but persistent high levels in dust. The findings highlight the need for improved dust management and cleaning practices to mitigate exposure risks and provide valuable insights for future intervention strategies in pig farming. I suggest taking it into consideration only after the answers to the raised queries and the modifications specified below:

Abstract: In the first part of the abstract quite a lot of general information is presented. Authors are kindly asked to shorten the abstract a little and to focus in particular on the important results obtained in the present study.

Introduction: The authors are advised to clarify the justification for focusing on the three sampling points (nasal swabs, air, and dust). They should provide more background on why nasal swabs and dust are particularly significant indicators of MRSA exposure risk.

L 43, 46: In the reviewer's opinion, far too many bibliographical sources are used in the mentioned paragraphs. The reviewer's suggestion is that the authors should revise the list of references and keep only the essential and suggestive references for the present article.

L 56, 58: The information in the mentioned lines fits the discussion section. The reviewer suggests reinserting them to the section mentioned above.

L 67-71: Same suggestion as above. The reviewer suggests to revise the Introduction and to make it more suggestive according to the suggestions.

L 101: “We monitored LA-MRSA levels” please try to avoid using personal style in the writing process of scientific articles. For academic style of writing, impersonal style should be used. Please rephrase and revise the whole manuscript according to this suggestion.

Materials and Methods: Why were nasal swabs, dust, and air chosen as the primary sample types? Were any other potential contamination routes (e.g., feed or water) considered for testing?

L 153, 160: I’m wondering, were all sampling batches collected in the same environmental conditions (e.g., temperature, humidity)? If not, could this have influenced the variation in LA-MRSA levels between batches?

L 154: Provide more information about the dust sampling process, including how representative the collected samples were and why certain sampling locations were chosen.

L 214: The results show significant batch-to-batch variation. What factors could explain this aspect, given that the pigs were treated identically? I’m wondering, could differences in farm management practices or ventilation systems have contributed to this variation? I suggest to expand the results and the discussion on batch variation regarded environmental or management factors that might contribute to LA-MRSA differences between batches.

L 326: Are there any specific recommendations for improving cleaning and disinfection routines in farms, particularly regarding reducing LA-MRSA in dust? How might the findings apply to practical farm management strategies? Offer clearer recommendations for reducing LA-MRSA in farm dust based on the study’s findings. Consider discussing more about the limitations of current disinfection methods.

L 357: From my point of view, the authors should mention all the limitations of the present study, if they exist, and of course, the perspectives and research topics for the future.

In my opinion, the manuscript entitled “Development of LA-MRSA Loads in Pigs and Pig Stables during the Fattening Period” present very interesting findings and it can be processed after minor correction and responses to the aforementioned queries and suggestions.

Author Response

The study entitled “Development of LA-MRSA Loads in Pigs and Pig Stables during the Fattening Period” investigates the development of livestock-associated methicillin-resistant Staphylococcus aureus (LA-MRSA) in pigs and their environment during the fattening period. Through the quantification of LA-MRSA in nasal swabs, air, and dust, the study reveals a significant reduction in nasal carriage over time, but persistent high levels in dust. The findings highlight the need for improved dust management and cleaning practices to mitigate exposure risks and provide valuable insights for future intervention strategies in pig farming. I suggest taking it into consideration only after the answers to the raised queries and the modifications specified below:

Abstract: In the first part of the abstract quite a lot of general information is presented. Authors are kindly asked to shorten the abstract a little and to focus in particular on the important results obtained in the present study.

We have shortened the abstract a little.

Introduction: The authors are advised to clarify the justification for focusing on the three sampling points (nasal swabs, air, and dust). They should provide more background on why nasal swabs and dust are particularly significant indicators of MRSA exposure risk.

We have now inserted a few lines about that at the end of the Introduction.

L 43, 46: In the reviewer's opinion, far too many bibliographical sources are used in the mentioned paragraphs. The reviewer's suggestion is that the authors should revise the list of references and keep only the essential and suggestive references for the present article.

We have removed one of the references but kept the rest since they deal with LA-MRSA from different angles or concern different methods used in the study.

L 56, 58: The information in the mentioned lines fits the discussion section. The reviewer suggests reinserting them to the section mentioned above.

L 67-71: Same suggestion as above. The reviewer suggests to revise the Introduction and to make it more suggestive according to the suggestions.

We have moved certain sections of the Introduction to the Discussion per request from other reviewers.

L 101: “We monitored LA-MRSA levels” please try to avoid using personal style in the writing process of scientific articles. For academic style of writing, impersonal style should be used. Please rephrase and revise the whole manuscript according to this suggestion.

This has now been rephrased.

Materials and Methods: Why were nasal swabs, dust, and air chosen as the primary sample types? Were any other potential contamination routes (e.g., feed or water) considered for testing?

These sample types were chosen because we know from previous studies that they are most often contaminated. Feed or water are not sources of MRSA. As also mentioned above, we have inserted a few lines in the introduction about this.

L 153, 160: I’m wondering, were all sampling batches collected in the same environmental conditions (e.g., temperature, humidity)? If not, could this have influenced the variation in LA-MRSA levels between batches?

The temperature in the barns is kept constant, but humidity may vary. However, we have no records on that, and to my knowledge, there are no references in the literature to this either. What we know is that LA-MRSA is very robust and can survive well for extended periods of time in manure and dust. This has also been addressed and referenced in the text.

L 154: Provide more information about the dust sampling process, including how representative the collected samples were and why certain sampling locations were chosen.

We have inserted a few lines about this now.

L 214: The results show significant batch-to-batch variation. What factors could explain this aspect, given that the pigs were treated identically? I’m wondering, could differences in farm management practices or ventilation systems have contributed to this variation? I suggest to expand the results and the discussion on batch variation regarded environmental or management factors that might contribute to LA-MRSA differences between batches.

All batches were raised under the exact same conditions concerning management, environment, feeding, etc., so we do not know what the reason for this batch-to-batch variation is. It was a surprise for us, too, and it has to our knowledge not previously been described. We have added a few lines in the Discussion on this.

L 326: Are there any specific recommendations for improving cleaning and disinfection routines in farms, particularly regarding reducing LA-MRSA in dust? How might the findings apply to practical farm management strategies? Offer clearer recommendations for reducing LA-MRSA in farm dust based on the study’s findings. Consider discussing more about the limitations of current disinfection methods.

We have inserted some text about national recommendations. However, they do not include recommendations for cleaning and disinfection.

L 357: From my point of view, the authors should mention all the limitations of the present study, if they exist, and of course, the perspectives and research topics for the future.

 We have now inserted a few lines on this at the end of the Discussion section. There is already some more in the Conclusion section.

In my opinion, the manuscript entitled “Development of LA-MRSA Loads in Pigs and Pig Stables during the Fattening Period” present very interesting findings and it can be processed after minor correction and responses to the aforementioned queries and suggestions.

Thank you.

Round 2

Reviewer 2 Report

Comments and Suggestions for Authors

I understand the limitations of the study and am satisfied with the response. The revised version of the study looks good and may be accepted. 

Comments on the Quality of English Language

Minor editing is required.

Author Response

We have now had someone proficient in English language (although not familiar with the topic of the paper) to go through the manuscript and we have revised accordingly. Hopefully, the manuscript is now improved in grammar, fluency and clarity.

Reviewer 4 Report

Comments and Suggestions for Authors

It is recommended to add the sentence Three of these strategies were tested twice, i.e., in two sections, and the last only once.” before “In the present investigation, we analyzed data from all control groups in those experiments.”

Author Response

We have inserted this sentence as requested.